# Cognitive ability and voting behaviour in the 2016 UK referendum on European Union membership

**Chris Dawson\*, Paul L. Baker**

School of Management, University of Bath, Bath, United Kingdom

\* c.g.dawson@bath.ac.uk

## Abstract

On June 23rd 2016 the UK voted to leave the European Union. The period leading up to the referendum was characterized by a significant volume of misinformation and disinformation. Existing literature has established the importance of cognitive ability in processing and discounting (mis/dis) information in decision making. We use a dataset of couples within households from a nationally representative UK survey to investigate the relationship between cognitive ability and the propensity to vote Leave / Remain in the 2016 UK referendum on European Union membership. We find that a one standard deviation increase in cognitive ability, all else being equal, increases the likelihood of a Remain vote by 9.7%. Similarly, we find that an increase in partner's cognitive ability further increases the respondent's likelihood of a Remain vote (7.6%). In a final test, restricting our analysis to couples who voted in a conflicting manner, we find that having a cognitive ability advantage over one's partner increases the likelihood of voting Remain (10.9%). An important question then becomes how to improve individual and household decision making in the face of increasing amounts of (mis/dis) information.

## Introduction

On June 23rd 2016 the UK voted to leave the European Union in a seemingly simple 'Remain in the European Union / Leave the European Union' referendum. In this referendum, voter turnout was 72.2% of the electorate with 51.9% voting to Leave and 48.1% voting to Remain [1]. The issue of Brexit was very controversial among the British public. Well-known celebrities such as Mick Jagger, Liz Hurley, Lord Ian Botham, Ringo Starr and Roger Daltrey supported the UK's exit, whilst a group of over 300 actors and writers, including J.K. Rowling and Benedict Cumberbatch wrote open letters urging a Remain vote. The issue was even divisive within the same political party. Then Prime Minister David Cameron was a Remain supporter, as was his successor Prime Minister Theresa May. However, Theresa May's resignation was followed by Prime Minister Boris Johnson who was a vocal Leave supporter as compared to his successor, Prime Minister Liz Truss who voted to Remain in the EU but subsequently expressed support for Brexit. Finally, the current Prime Minister Rishi Sunak voted to Leave

**Data Availability Statement:** The data underlying the results presented in the study are available from https://beta.ukdataservice.ac.uk/datacatalogue/series/series?id=2000053#!/access-data and can be found within the Access Data

dropdown menu with the following title: 6614
Understanding Society: Waves 1-12, 2009-2021
and Harmonised BHPS: Waves 1-18, 1991-2009.
Our complete analysis scripts and code book have
been posted at the Open Science Framework
(OSF): https://osf.io/3yn2b/.

**Funding:** The author(s) received no specific
funding for this work.

**Competing interests:** The authors have declared
that no competing interests exist.

the EU. Despite the mixed views of Britain's politicians and celebrities, the result is generally considered to have been a surprise with 10 Nobel-prize winning economists making the case in the days leading up to the vote for Remaining in the EU [2], and polling firms and book-makers predicting the Remain side's win [3–5]. On June 24th 2016 the British Pound Sterling and FTSE markets suffered one of their largest single day losses indicating that the referendum result was similarly a shock to the markets. Not surprisingly, the UK's withdrawal process from the EU has not been simple. The UK formerly began the withdrawal process in March 2017 and completed the negotiation process in March 2019 at which point it entered the transition phase which came to an end in December 2020. Over this complicated period, there have been suggestions that a substantial proportion of Leave supporters have since come to regret their vote [6,7].

Given the surprise result alongside the economic and political significance of the UK leaving the EU, a substantial and growing academic literature has emerged which seeks to understand the referendum result by analyzing the correlation between voter socioeconomic, sociodemographic, and psychological characteristics and a vote to Leave or Remain [3–5,8]. In contrast, the role of cognitive ability in explaining the referendum result has attracted little attention by academics. This is perhaps surprising given that the topic often arises in social media debates between Leave and Remain voters. Indeed, there are theoretical reasons to think that cognitive function may be associated with voting behavior in the referendum.

Our starting point is the burgeoning interest—largely as a consequence of the referendum and to some extent the U.S. elections in 2016—regarding individual vulnerability to misinformation (wrong, an accidental falsehood) and disinformation (knowingly wrong, a deliberate falsehood) [9]. The proper functioning of democracies requires that citizens are well informed about the decisions at stake [10]. Whilst political ignorance, or uninformedness may be harmless—as individual errors tend to cancel each other out in the electorate [11]—political action based upon misinformation can lead to substantial decision errors, distorting the outcomes of electoral processes [12]. At the extreme, wide-spread misinformation can lead to collective preferences that are far different from those that would exist if people were correctly informed [13]. The information provided to the public in the months leading up to the referendum on the UK's membership in the EU has been characterized as contradictory, false, and often fraudulent. This has been found to be especially true regarding the pro-Leave campaign [14,15]. In particular, Höller [14] found that misleading information was even shared by leading UK politicians, especially the politicians who were arguing the case for a vote to Leave the EU. In contrast, pro-Remain politicians analyzed in Höller [14] were found to stick more closely to the facts, although some of their points were found to be speculative. Additionally, recent work by Simpson and Startin [15] examined the influential role that the British Tabloid Press (BTP) had in shaping the pro-Leave campaign. In a content analysis of BTP headlines leading up to the referendum, Startin & Simpson [15] categorize almost all of the headlines as 'Questionable' or 'Speculative' with respect to their interpretation of underlying evidence or lack thereof. Indeed, Partheymüller, Kritzinger and Plescia [16] find that misinformedness is strongly associated with a preference to Leave the EU, while the well-informed prefer to Remain. This is consistent with van Kessel, Sajuria and van Hauwaert [17] who find that misinformedness is positively related to support for right-wing populist parties, and van Prooijen and Krouwel [18] who find that low knowledge tends to predict anti-establishment voting. In addition to the evidence of significant misinformation, the pro-Leave campaign has also been found to have dominated the day-to-day volume of tweets by a factor of four—especially in the three weeks leading up to the vote [19]. Whilst media outlets have always circulated some amounts of misleading information, the rise of social media and the internet has sharply increased the scale and accessibility of (mis/dis) information and of more divisive messages

[20]. As recently revealed by the whistleblower Frances Haugen, social media algorithms consciously privilege the most divisive content to amplify traffic on the networks [21]. This is consistent with the strong relationship between exposure to online political activity and Euroscepticism [21–23]. Interestingly, in Fortunato and Pecoraro [21] this relationship was only evident amongst individuals with relatively low levels of education. The argument here is that education captures an individual's cultural capital, that is, the knowledge, skills and experience that are accumulated over time [24]. Those with low education and therefore low cultural capital are more likely to be receptive to divisive ideas [25].

The literature on how cognitive function influences people's interaction with (mis/dis) information (see Pantazi, Hale and Klein [9] for a review), suggests that higher cognitive ability and analytical thinking are linked to an increased propensity to detect and resist misinformation [26–28]. Those with lower cognitive ability are also found to be less likely to adjust their judgment after they learn that important information on which their initial evaluation was based is incorrect [29] and are more susceptible to false memories arising from exposure to fabricated news stories [30]. Specific to Brexit, Greene, Nash and Murphy [30] also illustrate that individuals with better knowledge about the referendum showed better discrimination between true and false stories. Further empirical evidence on the link between false memories and cognition is provided by Murphy et al. [31] from the Irish abortion referendum. They find that those who reported voting in favor of legalizing abortion were more likely than "no" voters to "remember" a fabricated scandal about the campaign to vote "no", whilst "no" voters were more likely than "yes" voters to "remember" a fabricated scandal about the campaign to vote "yes." This relationship, where respondents formed false memories for fake news that aligned with their beliefs, was found to be particularly important for those low on cognitive ability.

Cognitive function may also be associated with voting behavior in the referendum indirectly, that is, via a third variable. The standard omitted variable bias problem. Here, the literature points to two potential candidates. Firstly, there is reason to believe that cognitive ability is associated with voting behaviour through its correlation with personality traits. There is a growing literature on the relationship between the Big Five personality traits (Openness, Conscientiousness, Extraversion, Agreeableness and Neuroticism) and attitudes toward the EU (see for e.g., Bakker and de Vreese [32] and Nielsen [33]). More specifically with respect to the referendum, recent work has shown that Openness (to new experiences) is correlated with support for Remaining in the EU [8,34] and Conscientiousness with support for Leaving the EU [34]. Here, Openness is relevant as it influences whether individuals see EU integration as an opportunity or threat. Alongside this research, within social psychology there exists a broad literature on the relationship between cognitive ability and personality, with studies finding a positive relationship between general intelligence and Openness (see Zeidner and Matthews [35] for a review as well as McCrae [36] and Curtis, Windsor and Soubelet [37]) and a negative relationship with Conscientiousness [38]. Intelligence has also been shown to have a positive effect (at the country level) on considering the interests of others [39]. Secondly, research has also found that lower cognitive ability predicted higher Right-Wing Authoritarian (RWA) scores [40,41] and a higher incidence of prejudice [42]. Specific to the EU, support for Brexit and fear about the EU has also been associated with authoritarian and nationalistic orientations [34], which has been found to be associated with less cognitive flexibility [43] and reduced strategic information processing [44].

A final channel through which cognitive function may be associated with voting behavior comes from the literature on behavioural biases. Here, the literature suggests that those with lower cognitive ability are more likely to succumb to judgemental biases when making decisions [45–48]. In this view, Sumner et al. [34], using the Cognitive Reflection Test (CRT)—which measures analytical thinking and the propensity to reflect on and override intuitive (but

incorrect) solutions—find that Leave voters were the most reliant on System 1 thinking—the fast, automatic, unconscious, and emotional decision system—rather than System 2 thinking —the analytical, deliberate and rational decision system. Indeed, for the substantial proportion of Leave supporters that have since come to regret their vote, the vote to Leave may be classified as a decision error.

Using a nationally representative sample of 6,366 individuals from 3,183 heterosexual couples from the UK Understanding Society panel survey and controlling for a complete set of socioeconomic and sociodemographic characteristics motivated by the previous literature, we investigate the relationship between cognitive ability—as measured by a broad range of cognitive skills, including memory, verbal fluency, fluid reasoning and numerical reasoning—and the propensity to vote Leave / Remain in the referendum. As our interest is in the direct relationship between cognitive ability and a vote to Leave / Remain as a function of processing referendum campaign (mis/dis) information, we include variables for the Big Five personality traits and political preferences to control for the potential indirect relationships discussed above. Our empirical strategy includes three parsimonious estimation techniques. In the first instance, treating individuals within couples as individual units, we estimate baseline coefficients for the impact of cognitive ability on voting behaviour. In the second stage of our analysis, we control for couple interdependence and couple characteristics on voting behaviour. In the final stage, as our strictest test, we control for household fixed-effects, that is, restricting all variation to within couples. This final test is important. For instance, cognitive ability is likely to be associated with heterogeneous exposure to information sets and/or experiences of living in the UK which are both likely to be highly informative for individual political decision making. However, within couples these differences are likely to be equalized, and, in the spirit of assortative mating, individuals within couples are likely to be similar to each other in other unobservable ways. Across all models, we find statistically and economically significant evidence that those with higher cognitive ability (and higher cognitive ability partners) are associated with a large increase in their propensity to vote Remain. Whilst this paper's focus is restricted to the UK's 2016 referendum on its membership in the European Union, our findings are important to the debate regarding the strengths (e.g., directly acting on the will of the people) and potential weaknesses (e.g., the ability of the electorate to decide a technically complex issue in isolation) of referendums more generally [49,50]. Our work is particularly important as referendums continue to be relevant instruments as exemplified by the (on average) four referendums per year that Switzerland holds as well as a potential Scottish Independence referendum.

Two other papers look at the relationship between cognitive ability and the referendum on the EU. Sumner et al. [34] is the closest paper to our work here. However, it is differentiated in that it is a broad study of personality traits, cognition biases and a single measure of cognitive ability (numeracy) and an individual's intention to vote Leave / Remain in the EU. Using a sample of participants recruited from Facebook advertising, Sumner et al [34] find that those who intended to vote Leave in the referendum had lower levels of numeracy. Carl [51] also focuses on voting intentions and finds a small positive correlation between average IQ and four measures of political attitudes, including voting intention in the referendum, at both the UK regional and local authority level. In contrast to this previous work, our paper is the first, as far as we are aware, to use a nationally representative micro level sample to focus on the relationship between individuals' cognitive ability and actual—rather than intended—voting behaviour in the referendum. This is an important distinction in work that (at least in part) seeks to explain the outcome of the referendum result. Furthermore, our paper uses five individual measures and one summative measure of cognitive ability which contributes to the comprehensiveness and robustness of our results. Finally, our paper is further distinguished

from Carl [51] which uses aggregate level data and is therefore potentially subject to the concern of ecological fallacy [3].

## Data and descriptive statistics

To understand the correlation between voting behavior in the referendum and cognitive ability, we use data from Understanding Society (USoc) 2009 to 2020 (Waves 1–12). USoc is a British, nationally representative annual longitudinal survey of approximately 40,000 households, funded by the UK Economic and Social Research Council. The survey instrument is a questionnaire involving a household section and individual sections, which cover a broad range of subjects including labor market activity, household dynamics, attitudes and opinions, amongst others. The analytic sample includes data from all households in which there was a heterosexual couple and who both gave valid responses to the dependent, independent and control variables used in the subsequent analysis. This yields a final analytic sample of 6,366 individual respondents from 3,183 heterosexual couples.

### Dependent variable

In Waves 10, 11 and 12 of USoc, participants were asked: "Did you vote in the EU referendum held on June 23rd, 2016?". Participants who reported voting were then asked: "How did you vote?" [Remain a member of the European Union; Leave the European Union]. Some respondents were asked this question in multiple Waves; for these people we recorded their first given response. A small sample of individuals report inconsistent responses to the question on voting behaviour in the referendum across the three Waves of data (i.e., Waves 10, 11 and 12). As a robustness check we delete these individuals from the sample. Our results with this slightly smaller sample are quantitively and qualitatively similar to our main analysis. Across the sample, 56.6% of our sample report voting to Remain in the EU. This is moderately higher than the outcome of the referendum on the EU (i.e., 48.1% to Remain). Responses within couples were highly correlated, with 85.5% of couples voting in the same direction in the referendum. This highlights the importance of using an analytic technique which can account for the relationship between respondent and partner responses.

### Independent variable

In Wave 3 of USoc, prior to the referendum held on June 23rd, 2016, cognitive function measures were collected through five cognitive tasks. The first task assessed respondents' memory, using an immediate and delayed word recall task. In short, participants were read a series of 10 words and were then asked to recall (immediately afterwards and then again later in the interview) as many words as possible, in any order. The scores from the immediate and delayed word recall task are then summed together resulting in a measure of cognitive function which we refer to as 'Word Recall'. The second task assessed semantic verbal fluency, where participants were given one minute to name as many animals as possible. The final score on this item is based upon the number of unique correct responses. This provides us with another measure of cognitive function which we refer to as 'Verbal Fluency'. The third task assessed respondents' working memory. Here, participants were asked to give the correct answer to a series of subtraction questions. There is a sequence of five subtractions, which started with the interviewer asking the respondent to subtract 7 from 100. The respondent is then asked to subtract 7 again, and so on. The number of correct responses out of a maximum of five was recorded, giving us a measure of cognitive function, referred to here as 'Subtraction Test'. The fourth task assessed respondents' fluid reasoning where participants were asked to write down a number sequence—as read by the interviewer—which consists of several numbers with a blank

number in the series. The respondent is asked which number goes in the blank. Participants were given two sets of three number sequences, where performance in the first set dictated the difficulty of the second set. The final score is based on the correct responses from the two sets of questions—whilst accounting for the difficulty level of the second set of problems—giving us a measure of cognitive function, 'Fluid Reasoning'. The final task assessed practical numerical knowledge. Participants were asked up to five questions that were graded in complexity. The type of questions asked included: "In a sale, a shop is selling all items at half price. Before the sale, a sofa costs £300. How much will it cost in the sale?" and "Let's say you have £200 in a savings account. The account earns ten percent interest each year. How much would you have in the account at the end of two years?". Based on performance on the first three items, participants are then asked either two additional (more difficult) questions or one additional (simpler) question. The final score is a count measure based on the number of correct responses. This provides us with our final measure of cognitive function referred to as 'Numerical Reasoning'. These measures of cognitive function have previously been successfully used in studies (for example) regarding COVID-19 vaccine hesitancy [52].

For the analysis that follows, we residualized the individual scores from each of the five cognitive tasks scores on a third-order polynomial of age. The procedure is necessary as cognitive function is highly dependent on age [53], and the vote on EU membership occurred several years after the elicitation of cognitive function. For completeness, we note that residualizing the cognitive function scores on age, whilst important, has little impact on the estimated coefficients in the subsequent analysis. From these residualized scores we also create a composite measure of cognitive function which we refer to as 'Cognitive Ability'. This composite measure standardizes and then combines the residualized scores from each of the five cognitive tasks. The level of internal consistency across the five items appears to be high: Cronbach's alpha is 0.71. Scores of 'Cognitive Ability' within couples were moderately correlated ($r = 0.317$, $p < 0.001$).

Fig 1 reports the bivariate relationship between our composite measure of cognitive ability, which is converted into deciles, and the frequency of voting Remain in the referendum. The figure shows that the frequency of voting Remain is monotonically increasing in the deciles of the cognitive ability distribution. Specifically, it shows that only 40% of people in the lowest decile of the cognitive ability distribution voted Remain, whereas approximately 73% of people in the highest decile of the cognitive ability distribution voted Remain.

In addition, Fig 2 reports the distributions of our composite measure of cognitive ability, which is standardized for ease of interpretation, for those who voted Remain or Leave in the referendum. The mean of the Leave (Remain) voters cognitive ability distribution is -0.245 (0.188) which suggests that the average Leave (Remain) voter falls below (above) the population average. The difference in means between voter types is 0.433 of a standard deviation, with a t-test confirming the difference is statistically significant, $t = 17.5$, $p < 0.001$. It is however important to recognize that there exists a significant overlap between the distributions of Remain and Leave voters cognitive abilities. Indeed, we calculated that approximately 36.3% of Leave voters had higher cognitive ability than the average (mean) Remain voter.

## Control variables

There are many factors that may predict voting behavior in the referendum. We included a comprehensive set of controls at the individual and household level that were measured in Wave 6 of USoc, just prior to the referendum. The individual level controls included age (in cubic form); gender; ethnicity, education; labour force status; interview mode; the number of sources used for information about news and current affairs; type of newspapers used for information about news and current affairs; political party supports/most aligned to; self-

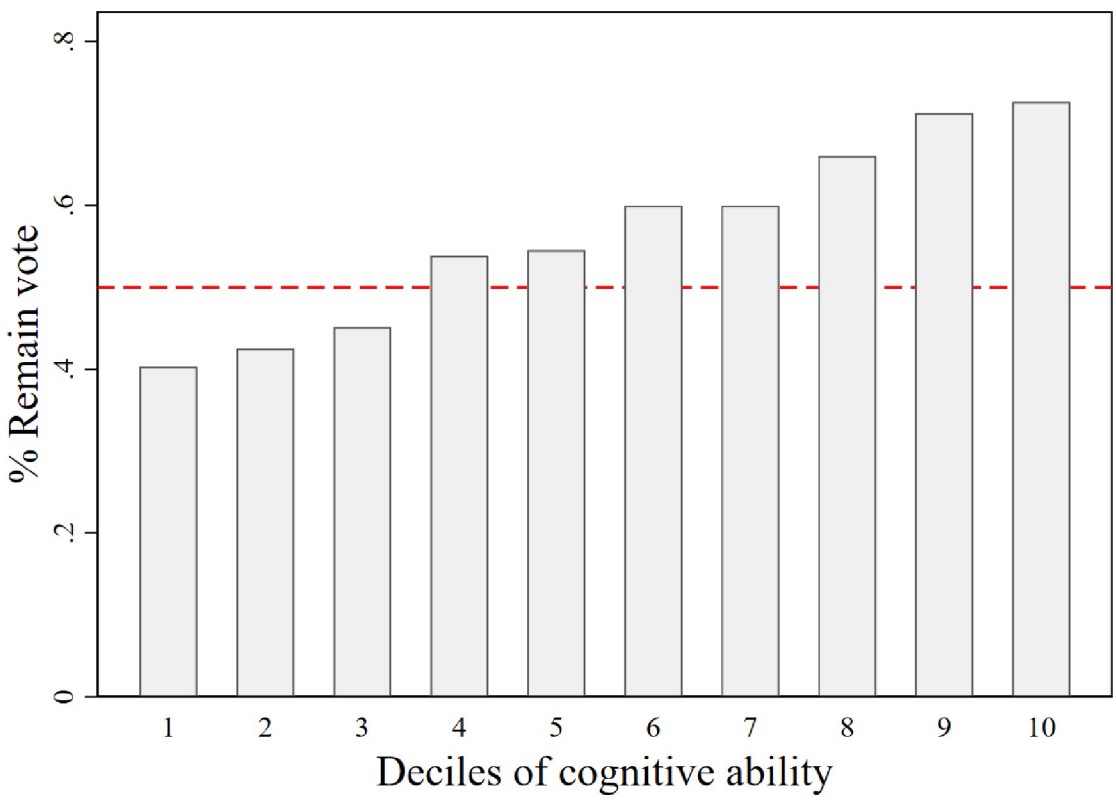

**Fig 1. Cognitive ability and a vote to remain. Bar chart of the relationship between deciles of our composite measure of cognitive function, 'Cognitive Ability', and the frequency of a Remain vote.** Horizontal dashed line at 50%. Sample of 6,366 individuals from 3,183 households.

assessed general health; whether the respondent suffers from long-term health problems; and personality traits—Openness, Neuroticism, Extraversion, Conscientiousness and Agreeableness—measured using the short 15-item Big-Five inventory (BFI-15). The household level controls included the logarithm of monthly household income (adjusted by the OECD-modified equivalence scale and deflated by the Consumer Price Index); marital status; the number of dependent children in the household; the square root of household size; housing tenure; whether the respondent or partner is the household financial decision maker; lives in an urban location; and a set of region of residence dummy variables. These factors have been shown to be strong predictors of voting behaviour [3–5,15].

Table 1 shows the correlations between the dependent variable, independent variable and a selection of the key control variables used in the study and S1 Table presents summary statistics. The mean age is approximately 54 years and just under 90% of couples were married. Exactly 36.6% reported having a university or college degree and 63.6% reported being an employee or in self-employment. Just over 15% of the sample do not support a political party, whilst 33.7% reported supporting The Conservatives, 26.9% supporting Labour, with just over 12% supporting nationalist political parties.

## Empirical analysis and results

### Main results

To determine our baseline estimated relationship between cognitive ability and voting behavior in the referendum, we treat our 6,366 individuals within 3,183 couples as individual units.

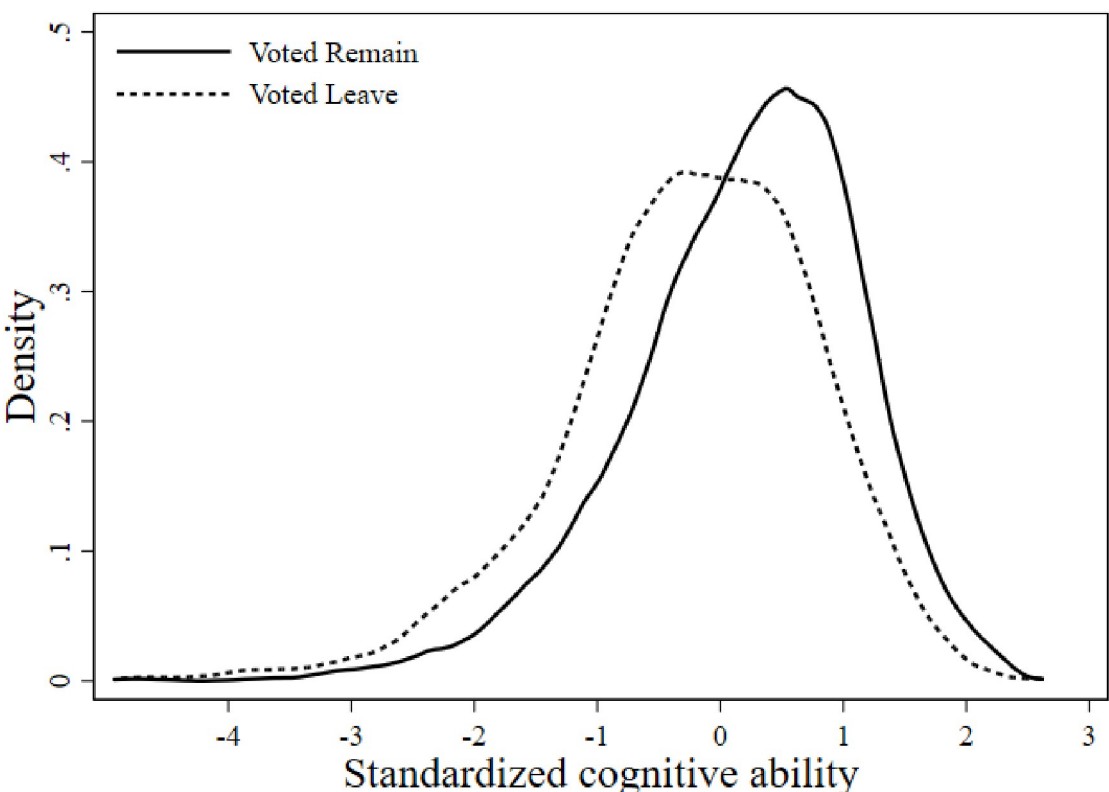

**Fig 2. Cognitive ability and a vote to remain or leave. Distributions of our standardized composite measure of cognitive function, 'Cognitive Ability', for Remain and Leave voters.** Sample of 6,366 individuals from 3,183 households.

**Table 1. Pairwise correlations.**

| Variables | (1) | (2) | (3) | (4) | (5) | (6) | (7) | (8) | (9) | (10) | (11) | (12) | (13) |
|---|---|---|---|---|---|---|---|---|---|---|---|---|---|
| (1) Voted Remain | | | | | | | | | | | | | |
| (2) Cognitive Ability | .215* | | | | | | | | | | | | |
| (3) Age (years) | -.159* | .000 | | | | | | | | | | | |
| (4) Male | -.035* | .128* | .080* | | | | | | | | | | |
| (5) White | -.088* | .129* | .116* | .018 | | | | | | | | | |
| (6) Education | .339* | .394* | -.281* | .025 | -.084* | | | | | | | | |
| (7) News sources | .134* | .142* | -.100* | .018 | -.019 | .215* | | | | | | | |
| (8) Household income | .226* | .241* | -.147* | .000 | .009 | .374* | .146* | | | | | | |
| (9) Broadsheets | .220* | .230* | .115* | .041* | -.017 | .301* | .316* | .217* | | | | | |
| (10) Redtop tabloids | -.126* | -.152* | -.032 | .062* | .041* | -.210* | .061* | -.125* | -.189* | | | | |
| (11) Conservative | -.116* | .036* | .141* | .033* | .075* | -.051* | -.020 | .085* | .022 | -.078* | | | |
| (12) Labour | .197* | .002 | -.061* | -.005 | -.097* | .096* | .090* | -.004 | .082* | .088* | -.432* | | |
| (13) Liberal | .140* | .089* | -.011 | -.027 | -.005 | .098* | .046* | .042* | .071* | -.071* | -.203* | -.172* | |
| (14) Nationalist | -.198* | -.077* | .009 | .046* | .028 | -.131* | -.064* | -.107* | -.115* | .066* | -.267* | -.227* | -.106* |

Notes: Sample of 6,366 individuals from 3,183 households. Cognitive ability is our composite measure of cognitive function. Education is educational attainment and is included as a 6-point scale from 1 = *No formal qualifications* to 6 = *University or college degree*. News sources is a count variable, which counts the number of media sources the respondent uses to get their information about news and current affairs. Household income is the logarithm of monthly household income which has been adjusted by the OECD-modified equivalence scale and deflated by the Consumer Price Index. Broadsheets and Redtop tabloids are binary variables indicating newspaper usage. Broadsheets are regarded as a more serious and less sensationalist outlet whilst redtop tabloids are regarded as sensationalist outlets which all have red mastheads. Conservative, Labour, Liberal and Nationalist represents the political party the respondent supports/most aligned to.
* $p < 0.01$.

As the dependent variable is binary, we estimate a pooled logistic regression, where we include as a key predictor our composite measure of cognitive function, 'Cognitive Ability', whilst controlling for our full set of control variables. Whilst our primary interest is our composite measure of cognitive ability, we also separately estimate the relationship between scores on each of the five cognitive function tasks and voting behaviour. This is to ascertain whether certain aspects of cognitive function may be more important than others in explaining voting behaviour. All cognitive function measures in the analysis that follows were standardized to ease interpretation of effect sizes. Column 1 of Table 2 reports the estimated coefficients. Specifically, each row reports the coefficients on the relevant measure of cognitive function from separate estimations. We report marginal effects and t-statistics, where the standard errors are clustered to control for intra household correlation. The standardized composite measure of cognitive function and the standardized scores on each of the five cognitive tasks are all highly correlated with voting to Remain a member of the EU. These effects are not small; for instance, a one standard deviation increase in 'Cognitive Ability' increases the likelihood of voting to Remain a member of the EU by 5.5 percentage points. Relative to the mean level of voting Remain in the sample, this relates to an increase of approximately 9.7%. It is also important to note, all aspects of cognitive function were separately important for voting behaviour in the referendum. Furthermore, we investigated whether the estimated coefficients presented in Column 1 were similar when we expanded our sample to include all respondents who were not part of a couple. This expands the sample size to 9,879 respondents. Here, the results are almost identical to those presented in Column 1, suggesting that there is nothing statistically unusual about our subsample of heterosexual couples.

We then repeat the analysis but now our focus is on the relationship between cognitive ability and voting behavior conditional on household effects. Initially, to account for the interdependence between members of a couple, we constructed multilevel logistic models (analogous to random-effects estimation) to estimate the actor-partner interdependence model [53]. Here, the model treats the actor as the Level 1 unit and the couple as the Level 2 unit. Next, using the actor-partner interdependence model, we estimated the extent to which spouses' cognitive ability predicted actors' voting behaviour above and beyond the actor effects. The estimated coefficients from these procedures are reported in Column's 2 and 3 of Table 2, respectively. We again report marginal effects and t-statistics, where the standard errors are clustered to control for intra household correlation. The results presented in Column 2 confirm our baseline estimates. Again, each row reports the coefficients on the relevant measure of cognitive function from separate estimations. The marginal effects are slightly smaller than those presented in Column 1; however, the effect sizes are still large and economically significant. In Column 3, where we include the partner's corresponding cognitive function score alongside the cognitive function score of the actor, we find that partner cognitive ability predicted voting behaviour of the actor. For instance, a one standard deviation increase in partner's 'Cognitive Ability' controlling for actors 'Cognitive Ability' increases the likelihood of the actor voting to Remain a member of the EU by 4.3 percentage points. Relative to the mean level of voting Remain in the sample, this relates to an increase of approximately 7.6%.

Whilst the previous multilevel strategies control for household random-effects, our third and final strategy controls for household fixed-effects. Specifically, using a conditional (fixed-effect) logistic estimator, we focus on the 926 respondents, from 463 couples where the actor and partner voted in a conflicting manner. This approach allows us to estimate the relationship between cognitive ability and voting behaviour purely from the variation that occurs within households. In Fig 3 we plotted a histogram of actor-partner differences in 'Cognitive Ability' to illustrate the substantial identifying variation for this final estimation approach, where the actor and partner voted in a conflicting manner. In the context of voting behaviour and

**Table 2. Pooled, multilevel and conditional (fixed-effect) logistic regressions measuring the relationship between cognitive ability and voting behaviour in the referendum.**

| | (1) | (2) | (3) | (4) |
|---|---|---|---|---|
| Dependent variable: | Voted Remain | Voted Remain | Voted Remain | Voted Remain |
| Regression: | Pooled logistic | Multilevel logistic | Multilevel logistic | Conditional logistic |
| *Actor*: | | | | |
| Cognitive Ability | 0.055*** | 0.037*** | 0.054*** | 0.109*** |
| | [6.376] | [5.567] | [7.495] | [2.600] |
| Word Recall | 0.033*** | 0.022*** | 0.036*** | 0.046 |
| | [4.291] | [3.748] | [5.383] | [1.223] |
| Verbal Fluency | 0.022*** | 0.011* | 0.020*** | -0.018 |
| | [2.759] | [1.659] | [2.983] | [-0.474] |
| Subtraction Test | 0.019** | 0.014** | 0.020*** | 0.048 |
| | [2.502] | [2.444] | [2.965] | [1.431] |
| Fluid Reasoning | 0.036*** | 0.026*** | 0.041*** | 0.120*** |
| | [4.446] | [4.240] | [5.692] | [3.021] |
| Numerical Reasoning | 0.044*** | 0.028*** | 0.046*** | 0.090** |
| | [5.351] | [4.198] | [6.360] | [2.164] |
| *Partner*: | | | | |
| Cognitive Ability | | | 0.043*** | |
| | | | [6.003] | |
| Word Recall | | | 0.031*** | |
| | | | [4.591] | |
| Verbal Fluency | | | 0.025*** | |
| | | | [3.742] | |
| Subtraction Test | | | 0.013* | |
| | | | [1.915] | |
| Fluid Reasoning | | | 0.029*** | |
| | | | [4.161] | |
| Numerical Reasoning | | | 0.041*** | |
| | | | [5.755] | |
| Number of individuals | 6,366 | 6,366 | 6,366 | 926 |
| Number of households | 3,183 | 3,183 | 3,183 | 463 |
| Additional controls | Yes | Yes | Yes | Yes |
| Household random effects | No | Yes | Yes | No |
| Household fixed effects | No | No | No | Yes |
| Mean dependent variable | 0.566 | 0.566 | 0.566 | 0.500 |

Notes: Main entries are marginal effects in Columns 1, 2 and 3, and average semi-elasticates in Column 4, t-statistics in square brackets. Standard errors are clustered at the household to control for intra household correlations. All columns include controls for age (cubic); gender; ethnicity, education; labour force status; interview mode; number of sources used for information about news and current affairs; type of newspaper used for information about news and current affairs; political party supports/most aligned to; self-assessed general health; whether respondent suffers from long term health problem; and personality traits—Openness, Neuroticism, Extraversion, Conscientiousness, Agreeableness—measured using the short 15-item Big-Five inventory (BFI-15). Columns 1, 2 and 3 include further controls for household specific factors including the logarithm of monthly household income (adjusted by OECD-modified equivalence scale and deflated by the Consumer Price Index); marital status; number of dependent children in the household; square root of household size; housing tenure; household financial decision maker; whether lives in urban location; and region of residence dummy variables. Full results presented in S2 Table of the Supporting Information. Significance levels *** 1%, ** 5%, * 10%.

cognitive ability, the introduction of household fixed-effects is important. For instance, it enables us to control for the possibility that variation in cognitive ability is strongly correlated with heterogeneous experiences of living in the UK, such as living in a neighbourhood with a

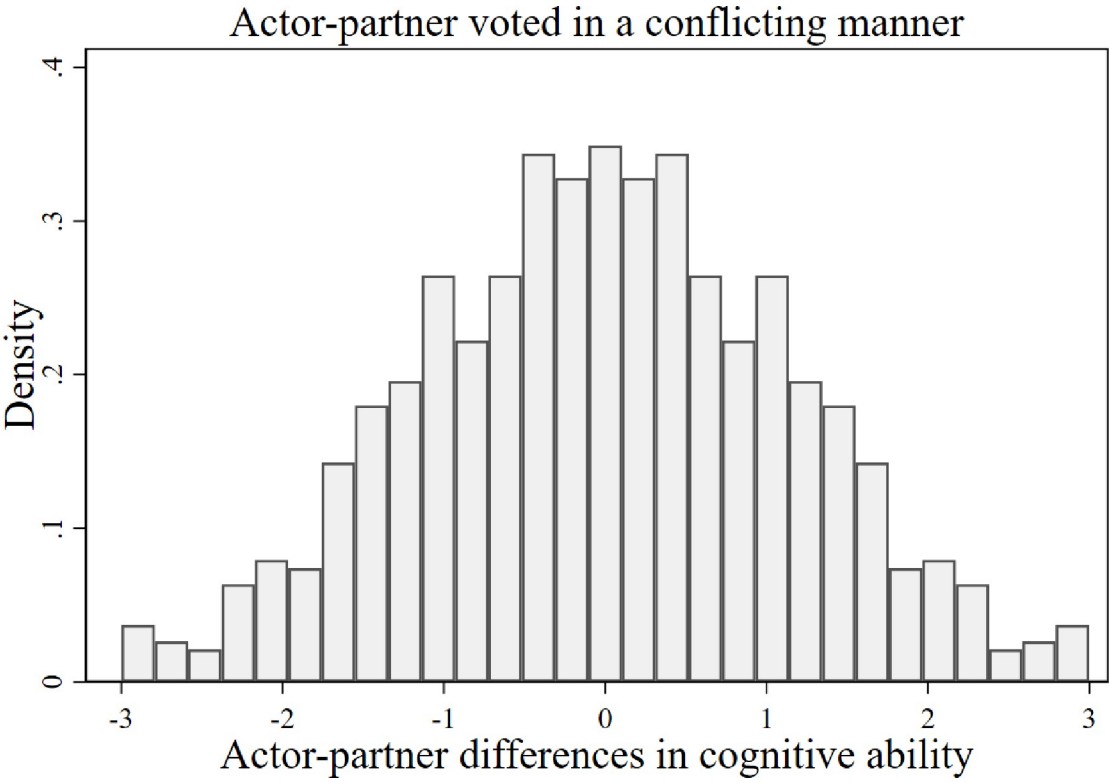

**Fig 3. Within household differences in cognitive ability. Histogram of the variation of our composite measure of cognitive function, 'Cognitive Ability', within households where the actor and partner voted in a conflicting manner.** Based on 926 individual observations from 463 households.

high concentration of EU migrants [54,55], which may be important for voting behaviour. More than this, variation in cognitive ability may also be strongly correlated with heterogeneous exposure to misinformation and disinformation, which has been shown to be correlated with voting behaviour in the referendum [16]. However, individuals within couples are likely to share and debate their information sets, and therefore by controlling for household fixed-effects we can control for information (dis)advantages. In short, we compare the effects of cognitive ability on voting behaviour for individuals exposed to the same contextual factors and information sets. Household variation in cognitive ability and voting behaviour is therefore likely to expose variation in how individuals process information sets [26–28] and interact and interpret their geographical context [54]. Column 4 of Table 2 reports the estimated coefficients of this procedure. Again, each row reports the coefficients on the relevant measure of cognitive function from separate estimations. We report average semi-elasticities for our conditional (fixed-effect) logistic approach and t-statistics, where the standard errors are clustered to control for intra household correlation. Average semi-elasticities are calculated as the conditional (fixed-effects) logistic regression does not produce estimates of average partial effects or marginal effects. Our results continue to hold. For instance, a one standard deviation 'Cognitive Ability' advantage over one's spouse increases the likelihood of voting to Remain in the EU by 10.9%. Whilst the memory task, 'Word Recall', 'Verbal Fluency' and the 'Subtraction Test' measures of cognitive function are no longer statistically significant at the conventional levels, we find highly statistically significant effects for the other measures of cognitive function. That is, 'Numerical Reasoning'—processing numerical patterns logically—and 'Fluid

Reasoning' which measures the capacity to think logically and solve problems in novel situations.

We briefly describe other significant Remain effects, revealed in the other covariate coefficient estimates that are incidental to the main themes of the paper. These full covariate estimates are reported in S2 Table. Most notably, from Column 1 and relative to not being a political supporter, supporting Labour, Greens and other liberal political parties increases the chances of voting Remain by 21.2, 19.3 and 25.2 percentage points, respectively. Conversely, and consistent with the previous literature, supporting nationalist parties reduced the probability of voting Remain by 24 percentage points [43]. These results were robust to the inclusion of household fixed-effects (Column 4 of S2 Table). The newspapers people use for their information about news and current affairs, consistent with previous research, also had a large effect on voting behavior as evidenced in Column 1 of S2 Table [15]. Specifically, relative to those who don't read newspapers, the tabloid newspapers—regarded as sensationalist outlets—reduced the probability of voting Remain by between 14 and 20 percentage points, depending on the type of tabloid newspaper. Consistent with the previous literature [56], higher levels of education are associated with an increased likelihood of voting Remain. In Column 1 of S2 Table—and to put the effect size of education in the context of cognitive ability—a movement in 'Cognitive Ability' from minus to plus one standard deviation from the mean has the same effect on the probability of voting Remain as having a university or college degree relative to having no formal education. In short, both education and cognitive ability mattered for the referendum result. This is additionally interesting as educational attainment and cognitive ability are fundamentally interconnected [57], therefore our results suggest these collinear variables do not contain the same information about the dependent variable. Education is an important control owing to the tendency for lower cognitive ability to be associated with lower levels of education, which is associated—perhaps because less educated voters are those who might find it harder to understand the opportunities from globalization that accompanies EU membership—with the tendency to vote Leave [3]. Whilst we control for educational attainment, it is possible that the type of university attended or the subject of study may be correlated with both cognitive ability and a tendency to vote Remain, leading to may a bias in our parameter estimates. Nevertheless, even after controlling for educational attainment and a host of other covariates, the observed positive relationship between cognitive ability and a Remain vote is extremely strong. This suggests that the likelihood of an unobservable confounder, such as the subject of study, accounting for the entirety of the estimated relationship between cognitive ability and a vote to Remain in the EU is rather low. Interestingly, the effect of educational attainment on voting behaviour was not robust to the inclusion of household fixed-effects (Column 4 of S2 Table), possibly because within households the cultural capital differences associated with variations in education disappear [24]. In terms of personality traits, from Column 1 of S2 Table, higher Neuroticism (emotional instability) and Agreeableness were found to increase the probability of voting Remain, although these effects were not robust to the inclusion of household fixed-effects (Column 4 of S2 Table). This is consistent with the hypothesized relationships in Sumner et al. [34].

## Robustness

Whilst the previous section illustrates that the relationship between cognitive ability and voting behaviour is robust to various estimation strategies, in this section, we perform additional checks. Firstly, we check for non-linearities in the relationship between cognitive ability and voting Remain. Secondly, we test whether our results derived from binary choice models are robust to the use of linear probability models, which then allows us to implement our final

strategy, which is to examine how our cognitive ability coefficients behave when the regression specification is modified by adding or removing regressors.

To investigate for non-linearities in the relationship between cognitive ability and voting Remain, we repeat the analysis in Table 2 and replace the linear composite measure of cognitive ability with indicator variables for being in each decile of the cognitive ability distribution, with the 1st decile being the omitted category. We plot the point estimates and 95% and 90% confidence intervals in Fig 4. Panels A, B, C, D of Fig 4 relate to the econometric specifications in Column's 1, 2, 3, 4 of Table 2, respectively. The effect of cognitive ability is approximately linear in all Panels of Fig 4.

Next, we repeat the analysis in Table 2, but instead of using binary choice models we use linear probability models. The results are described in S3 Table, which reassuringly reports the

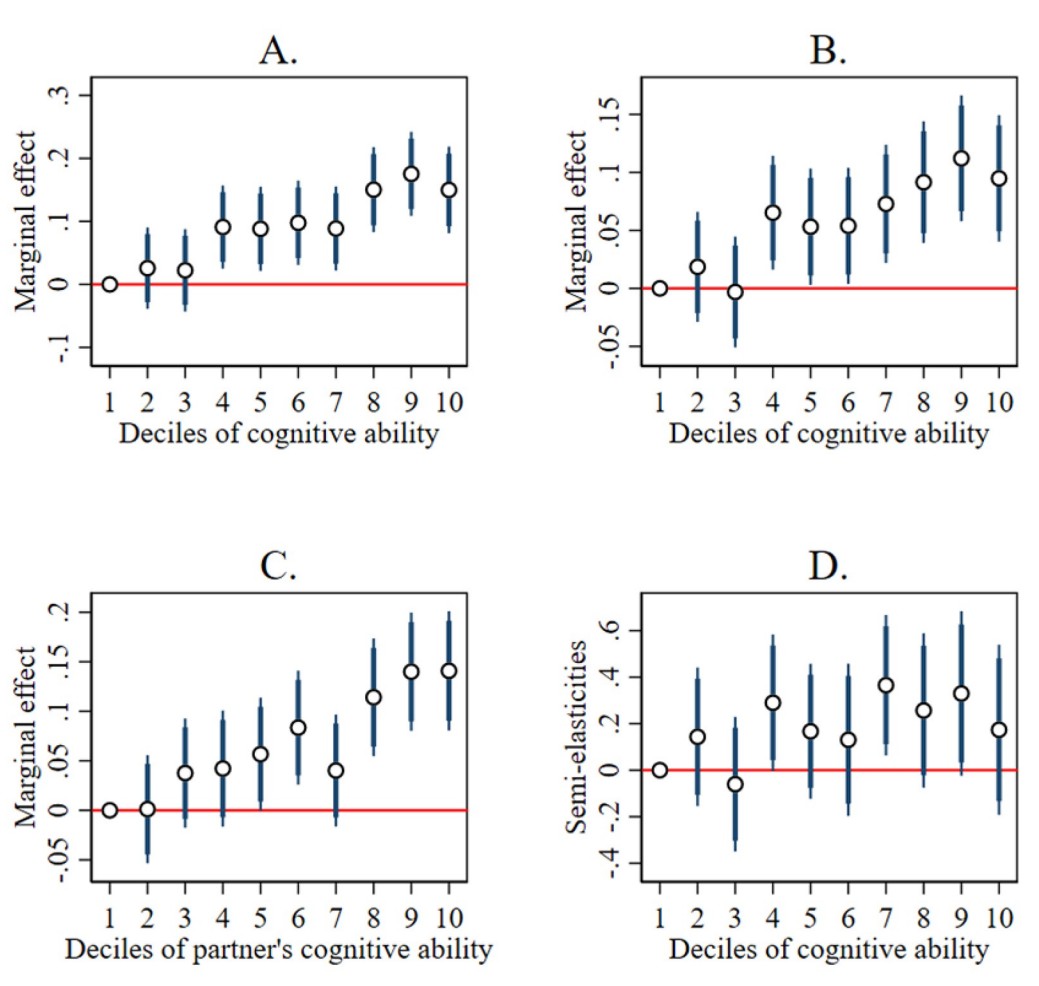

**Fig 4. Cognitive ability and the propensity to vote remain.** Estimates from regressions where our composite measure of cognitive function, 'Cognitive Ability', is entered in deciles. The omitted category is the 1st decile. Point estimates and 95% and 90% confidence intervals are shown. Panels A, B, C, D relate to the econometric specifications in Column's 1, 2, 3, 4 of Table 2, respectively.

same pattern of results established in Table 2. Lastly, we examine how our cognitive ability coefficients in S3 Table behave when the regression specification is modified by adding or removing regressors. Here, we adopt the approach proposed by Young and Holsteen [58], where our analysis is conducted across 128 unique combinations of control variables. S4 Table reports regressions with the same specifications as S3 Table, but where the reporting of coefficients reflects the mean estimate from the 128 unique combinations. The results of this procedure suggest our main findings are robust to combinations of control variables, and therefore do not depend on knife-edge specifications (i.e., a specific combination of regressors). Interestingly, the procedure developed in Young and Holsteen [58] also allows researchers to analyse how the introduction of a certain control variable changes the coefficient of interest. In all specifications, the most influential controls regarding the magnitude of the cognitive ability coefficients are: education; the number of sources used for information about news and current affairs; type of newspaper used for information about news and current affairs and political party supports/most aligned to. The introduction of these influential controls all serve to substantially lower the cognitive ability coefficients.

## Conclusion

This paper, using nationally representative data and a barrage of control variables, assessed the link between cognitive function and voting behavior in the referendum. Three key findings are produced. Firstly, cognitive ability mattered for voting behavior in the UK's 2016 referendum on its membership in the EU. Secondly, couples within households are highly interdependent, so much so, that spouse's cognitive ability is found to be strongly related to the agent's voting behaviour above and beyond the effect of the agents' own cognitive function. This is important as research typically considers political behaviour as an individual-level variable and has largely ignored the role of partners in shaping behaviour. Thirdly, when individuals within couples vote in a conflicting manner, cognitive ability remains an important factor in explaining referendum voting behavior. Specifically, having a one standard deviation cognitive ability advantage over one's spouse increases the likelihood of voting to Remain in the EU by 10.7%.

Taken together, these findings challenge the ideas that political behaviours are solely a function of socioeconomic issues. The idea that cognitive ability mattered for the referendum is consistent with the literature on cognitive ability and vulnerability to (mis/dis) information. It is suggestive that erroneous reporting surrounding the referendum might have complicated personal decision making, especially for those with low cognitive ability. It is also possible that those with lower levels of cognitive ability—rather than just lower levels of education—are more receptive to divisive ideas [25], or that that those lower in cognitive ability had less knowledge about the referendum which tends to predict anti-establishment voting [18]. On this later point, Carl, Richards and Heath [59] find those who score higher on probability reasoning also score higher on EU knowledge. These results do present some future challenges, most noticeably, how to improve decision making in the face of increasingly unmanageable amounts of (mis/dis) information, and more generally, the impact of (mis/dis) information on democratic processes. Of course, political action based upon misinformation can lead to substantial decision errors, distorting the outcomes of electoral processes by the 'active misinformed' [12]. If those lower in cognitively ability are more vulnerable to misinformation, then political campaigns based on (mis/dis) information may prevail depending on the ability distribution of the electorate. This is what Brennan [60] describes as the 'competence principle', which categorizes political decisions as unjust if they are made by a generally incompetent decision-making body.

It is possible the adverse effects of misinformation may be mitigated by those with lower cognitive ability having a lower propensity to turnout and vote [61]. However, in unreported

results, we fail to find evidence of a significant relationship between cognitive function and the likelihood of voting in the referendum on the EU. This may be influenced in some way by the particularly high voter turnout—72.2% of the electorate. This gives the rather severe option to democratic countries—who often limit or deny the voting rights of its citizens based upon cognitive impairments [62]—of restricting voting based on cognitive function. Perhaps though, a less severe restriction, especially when it comes to complex decisions like EU membership, is leaving it to the experts.

Lastly, there are, however, some limitations to our study. Most noticeably, the positive correlation between cognitive ability and voting to Remain in the referendum could, as always, be explained by omitted variable bias. Although we control for political beliefs and alliances, personality traits, a barrage of other socioeconomics factors and in our preferred model, household fixed-effects, the variation of cognitive ability within households could be correlated with other unobservable traits, attitudes and behaviours. The example which comes to mind is an individual's trust in politicians and government. Then Prime Minister of the UK David Cameron publicly declared his support for remining in the EU, as did the Chancellor of the Exchequer. The UK Treasury published an analysis to warn voters that the UK would be permanently poorer if it left the EU [63]. In addition to this were the 10 Nobel-prize winning economists making the case in the days leading up to the referendum. Whilst cognitive ability has been linked with thinking like an economist [64,65], Carl [51] also finds evidence of a moderate positive correlation between trust in experts and IQ. Moreover, work on political attitudes and the referendum have shown that a lack of trust in politicians and the government is associated with a vote to Leave the EU [56]. Therefore, the positive relationship between cognitive ability and voting Remain could be attributable for those higher in cognitive function to place a greater weight on the opinion of experts. A final note is that our dependent variable is self-reported which may induce bias, for instance, social desirability bias. Against that, the majority (75.6%) of responses were recorded through a self-completion online survey and we do control for interview mode, which produces no statistically significant effects. It is also important to recognise that this study is specific to analyzing a referendum. There is a political science literature that makes important distinctions between voting behavior in referendums versus elections [66]. Therefore, a fruitful area of future research would be to develop the ideas in this paper to the setting of election outcomes.

## Supporting information

**S1 Table. Descriptive statistics.**
(DOCX)

**S2 Table. Pooled, multilevel and conditional (fixed-effect) logistic regressions measuring the relationship between cognitive ability and voting behaviour in the referendum.**
(DOCX)

**S3 Table. Pooled, multilevel and fixed-effect linear regressions measuring the relationship between cognitive ability and voting behaviour in the referendum.**
(DOCX)

**S4 Table. Multiverse estimation: Pooled, multilevel and fixed-effect linear regressions measuring the relationship between cognitive ability and voting behaviour in the referendum.**
(DOCX)

## Acknowledgments

The authors would like to thank Chris Dimos, Sam Johnson, Thomas Roulet and Phil Tomlinson for helpful comments.

## Author Contributions

**Conceptualization:** Chris Dawson.

**Data curation:** Chris Dawson.

**Formal analysis:** Chris Dawson.

**Investigation:** Chris Dawson, Paul L. Baker.

**Methodology:** Chris Dawson, Paul L. Baker.

**Project administration:** Chris Dawson, Paul L. Baker.

**Writing – original draft:** Chris Dawson, Paul L. Baker.

**Writing – review & editing:** Chris Dawson, Paul L. Baker.

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
