## [Decision Letter · Decision Letter 0]

8 May 2023

PONE-D-23-06300Cognitive Ability and Voting Behaviour in the 2016 UK Referendum on European Union MembershipPLOS ONE

Dear Dr. Baker,

Thank you for submitting your manuscript to PLOS ONE. After careful consideration, we feel that it has merit but does not fully meet PLOS ONE’s publication criteria as it currently stands. Therefore, we invite you to submit a revised version of the manuscript that addresses the points raised during the review process.

Major Revisions required

We look forward to receiving your revised manuscript.

Kind regards,

Verda Salman, PhD

Academic Editor

PLOS ONE

Journal Requirements:

2. If published, this research might receive media attention, or may be press released by the authors or PLOS ONE. If you are considering this manuscript further for publication, we strongly suggest that you obtain evaluations from at least two external reviewers. If you feel the manuscript should be rejected without review, please clearly state your reasons for the decision in your letter to the authors, as is the requirement for all rejections. We also welcome your thoughts about whether this submission is press-worthy, which you may include in the confidential “Editor Comments to Editor” upon issuing a decision, or include in an email to plosone@plos.org. If you would like to discuss this work, or if you would like suggestions of peer reviewers, please email plosone@plos.org.

Additional Editor Comments:

Major Revisions are required

Reviewers' comments:

Reviewer's Responses to Questions

**Comments to the Author**

1. Is the manuscript technically sound, and do the data support the conclusions?

Reviewer #1: Yes

Reviewer #2: Yes

2. Has the statistical analysis been performed appropriately and rigorously? 

Reviewer #1: Yes

Reviewer #2: Yes

3. Have the authors made all data underlying the findings in their manuscript fully available?

Reviewer #1: Yes

Reviewer #2: Yes

4. Is the manuscript presented in an intelligible fashion and written in standard English?

Reviewer #1: Yes

Reviewer #2: Yes

5. Review Comments to the Author

Reviewer #1: 1. It's 2023 now. What's the point of the author studying the UK's vote to leave the European Union in 2016? What are the effects of these conclusions and research findings on the present?

2. In the fifth line of the second paragraph of the introduction, more than three authors, just list the first author and add et al. There are similar problems in the content behind the manuscript, the author himself checked it.

3. At the end of the introduction, the author puts forward the innovation of the manuscript, but the innovation is too little, so it is suggested to supplement it.

4. The footnote reference format of 4.2.1 is not correct.

5. 5.2.2 Whether the title can be considered as the independent variable.

6. The author's empirical results have achieved relatively good results, but there is no robustness test, and a robustness test needs to be added to prove that the results obtained by the author are robust.

7. The representation of the percentage in the manuscript is recommended to be unified. In some places, the symbol “%” is used, and in some places, the “percentage points” is used.

Reviewer #2: The paper titled; “Cognitive Ability and Voting Behaviour in the 2016 UK Referendum on European Union Membership” uses nationally representative data and a barrage of control variables to assess the link between cognitive function and voting behavior in the UK’s 2016 referendum on its membership in the EU. The findings are plausible and challenge the ideas that political behaviors of voters are solely a function of socioeconomic issues. The analysis is based on the premise that the period leading up to the referendum was characterized by a significant volume of misinformation and disinformation and it may affect the voter’s behavior. Moreover, it is obvious that higher cognitive ability and analytical thinking increase the propensity to detect and resist misinformation. Below are some of my observations. I could not understand why the author used cognitive ability and its constructs in the regression equations simultaneously. Would it not be causing multicollinearity problem? Moreover, I think cognitive ability measures may be highly subject to biasedness due to education level and discipline of the respondent. The results discussion is too brief and can be elaborated to make it clearer for the reader. It may improve the usefulness of the findings considerably.

6. PLOS authors have the option to publish the peer review history of their article (what does this mean?). If published, this will include your full peer review and any attached files.

Reviewer #1: No

Reviewer #2: **Yes: **Faisal Jamil

---

## [Author Response · Author response to Decision Letter 0]

5 Jun 2023

Please see the submitted Response to Reviewers Letter for detailed responses to all of the comments.

---

## [Decision Letter · Decision Letter 1]

5 Oct 2023

PONE-D-23-06300R1Cognitive ability and voting behaviour in the 2016 UK referendum on European Union membershipPLOS ONE

Dear Dr. Baker,

Thank you for submitting your manuscript to PLOS ONE. After careful consideration, we feel that it has merit but does not fully meet PLOS ONE’s publication criteria as it currently stands. Therefore, we invite you to submit a revised version of the manuscript that addresses the points raised during the review process.

Minor Revisions Required

We look forward to receiving your revised manuscript.

Kind regards,

Verda Salman, PhD

Academic Editor

PLOS ONE

Journal Requirements:

Additional Editor Comments :

Accepted for Publication

Minor Revisions Required

Reviewers' comments:

Reviewer's Responses to Questions

**Comments to the Author**

1. If the authors have adequately addressed your comments raised in a previous round of review and you feel that this manuscript is now acceptable for publication, you may indicate that here to bypass the “Comments to the Author” section, enter your conflict of interest statement in the “Confidential to Editor” section, and submit your "Accept" recommendation.

Reviewer #1: All comments have been addressed

Reviewer #2: All comments have been addressed

Reviewer #3: (No Response)

Reviewer #4: All comments have been addressed

2. Is the manuscript technically sound, and do the data support the conclusions?

Reviewer #1: Yes

Reviewer #2: Yes

Reviewer #3: Yes

Reviewer #4: Yes

3. Has the statistical analysis been performed appropriately and rigorously? 

Reviewer #1: Yes

Reviewer #2: Yes

Reviewer #3: Yes

Reviewer #4: Yes

4. Have the authors made all data underlying the findings in their manuscript fully available?

Reviewer #1: Yes

Reviewer #2: Yes

Reviewer #3: Yes

Reviewer #4: Yes

5. Is the manuscript presented in an intelligible fashion and written in standard English?

Reviewer #1: Yes

Reviewer #2: Yes

Reviewer #3: Yes

Reviewer #4: Yes

6. Review Comments to the Author

Reviewer #1: The author has been modified according to the opinions put forward, and has answered it very carefully. Finally, it is suggested that the limitations and future prospects of the study should be added at the end of the article.

Reviewer #2: The manuscript has been improved significantly and allthe comments are incorporated to my satisfaction.

Reviewer #3: Thanks, interesting manuscript. I was invited to the second round of reviews.

This is a professionally prepared manuscript, the methods are good and the topic is relevant.

I only have three smaller but important suggestions, all of them should be implemented. Plus some minor.

(1) If I put it polemically: The message of the study is that a) being for Brexit is stupid and that b) the people who voted for Brexit are stupid. Stupid + stupid = super stupid. It also becomes clear what opinion the authors themselves represent. To moderate the whole thing rhetorically, I would mention a few names of prominent people who were in favor of Brexit, right at the beginning. For example: “The issue of Brexit was very controversial among the British public. Well-known celebrities like Mick Jagger (+ more names) were for it, others (a few names) were against it.”

(2) Absolutely: Present a correlation table with correlations (r between –1, 0 and +1) between all important variables, most importantly: between intelligence and Brexit voting behavior.

(3) Add the IQ scores of voters for and against Brexit and the IQ difference. If you don't have these numbers, then on a small z scale. What is the difference between the two in SD units?

Abstract:

Add for this sentence ”We find that higher cognitive ability is strongly linked to a preference for Remain at the individual level ([Pearson/Spearman] r=.??), that partner’s cognitive ability influences the respondent’s preferences ([standardized] beta=.??) and lastly, that having a cognitive ability advantage over one’s partner increases the likelihood of voting Remain ([standardized] beta=.??).”

Introduction:

Intelligence also has a positive effect (at the country level) on considering the interests of others:

Rindermann, H. & Carl, N. (2020). The good country index, cognitive ability and culture. Comparative Sociology, 19, 39–68.

Results:

Figure 1 would be also possible as a scatterplot? (Not only bar chart) Add the Pearson/Spearman correlation in the notes of Figure 1.

Good luck, this study will be well received.

Reviewer #4: The author has responded to all of the feedback provided by reviewers. This paper is now ready for publication.

7. PLOS authors have the option to publish the peer review history of their article (what does this mean?). If published, this will include your full peer review and any attached files.

Reviewer #1: No

Reviewer #2: No

Reviewer #3: No

Reviewer #4: No

---

## [Author Response · Author response to Decision Letter 1]

13 Oct 2023

Please see the uploaded Response to Reviewers document.

---

## [Editor Report · Decision Letter 2]

23 Oct 2023

Cognitive ability and voting behaviour in the 2016 UK referendum on European Union membership

PONE-D-23-06300R2

Dear Dr. Baker,

We’re pleased to inform you that your manuscript has been judged scientifically suitable for publication and will be formally accepted for publication once it meets all outstanding technical requirements.

Kind regards,

Verda Salman, PhD

Academic Editor

PLOS ONE

Additional Editor Comments (optional):

Accepted for publication
---

## [Editor Report · Acceptance letter]

19 Jul 2023

PONE-D-23-06300R1 

Cognitive ability and voting behaviour in the 2016 UK referendum on European Union membership 

Dear Dr. Baker:

I'm pleased to inform you that your manuscript has been deemed suitable for publication in PLOS ONE. Congratulations! Your manuscript is now with our production department. 

Kind regards, 

on behalf of

Dr. Verda Salman 

Academic Editor

PLOS ONE